# Feasibility of Support by Family Practitioners in Reducing Colorectal Cancer-Related Death among Outpatients Who Have Not Undergone Colorectal Cancer Screening

**DOI:** 10.3390/diagnostics12081782

**Published:** 2022-07-22

**Authors:** Ryo Sugaya, Takeshi Kanno, Hirohisa Yasaka, Misuzu Masu, Masataka Otomo, Tomoyuki Koike

**Affiliations:** 1Division of Internal Medicine, National Health Insurance Marumori Hospital, Marumori 981-2152, Japan; m08051rs@jichi.ac.jp (R.S.); m10102hy@jichi.ac.jp (H.Y.); m-outomo@town.marumori.miyagi.jp (M.O.); 2Japan National Health Insurance Kuriyama Clinic, Nikko 321-2713, Japan; 3Department of Education and Support for Regional Medicine, Tohoku University Hospital, Sendai 980-8574, Japan; 4Division of Gastroenterology, Tohoku University Graduate School of Medicine, Sendai 980-8574, Japan; tkoike@rd5.so-net.ne.jp; 5Division of Internal Medicine, Medical Court Hachinohe West Hospital, Aomori 039-1103, Japan; m06024mo@jichi.ac.jp

**Keywords:** fecal occult blood test, colorectal cancer, cancer screening, reason for untested, primary care

## Abstract

We aimed to clarify the effectiveness of interventions in outpatients who did not undergo colorectal cancer (CRC) screening. From September 2012 to August 2013, we conducted a project, which showed that the immunological fecal occult blood test (FOBT) was actively recommended for outpatients who were ≥40 years of age, attended the Marumori Hospital regularly, and were not screened for CRC in the previous two years. We evaluated the detection rate of CRC and the disease specific survival ratio in February 2021 among patients with positive FOBT results during the retrospective cohort study. Overall, 388 (91%) out of the 425 outpatients submitted their stool samples. Among 388 outpatients, 66 tested positive for FOBT. While both the positive rate of FOBT and the detection rate of CRC (17% and 0.77%, respectively) were significantly higher than those in the nationwide administrative examination (5.7% and 0.13%, respectively) (*p* < 0.05), there was no statistically significant difference in the detection rate, compared with the group aged 65 years and older in the nationwide administrative examination. The 7-year CRC-specific survival ratio was 98.5%. Active promotion of FOBT at primary care institutions for outpatients who did not undergo CRC screening may contribute to reducing the frequency of CRC-related deaths.

## 1. Introduction

The incidence of colorectal cancer (CRC) is increasing worldwide and Japan is no exception, with CRC being the most common cancer in 2018 and the second-leading cause of cancer-related deaths in 2020 [1,2]. Screening examinations are widely recognized to play a significant role in the early detection of CRC [3,4]. A systematic review of four randomized clinical trials demonstrated that screening with the immunological fecal occult blood test (FOBT) reduces the risk of death from CRC by 18% (risk ratio, 0.82; 95% confidence interval (CI), 0.73−0.92) [5].

In Japan, although FOBT is recommended as a form of population-based screening for CRC among people aged ≥40 years, only 41.2% of these individuals underwent FOBT nationwide in 2019 [6]. The low rates of mass screening are considered problematic both in Japan and worldwide. Studies from outside Japan reported the characteristics of patients who do not undergo CRC screening, while Japanese studies have analyzed the factors encouraging people to undergo screening and the reasons for not undergoing secondary testing after screening [7,8,9,10,11]. However, few studies have attempted to evaluate the reasons for not undergoing screening among the roughly 60% of the Japanese population that is outside the scope of CRC screening projects. To limit the number of CRC-related deaths, medical professionals must determine the circumstances of people who go unscreened and the effects of interventions.

To address this objective, we developed a project at Marumori Hospital, a regional primary care hospital in Marumori, Miyagi (population, 15,501; aging rate ≥ 65 years, 33% as of 2010) [12]; we proactively recommended FOBT for regular outpatients and conducted secondary testing for patients with a positive FOBT result. In the present study, we aimed to determine the following: the percentage of patients who submitted samples for FOBT, characteristics of patients who tested positive in FOBT, secondary test results, and long-term outcomes.

## 2. Materials and Methods

### 2.1. Participants

We examined CRC screening records of the previous two years for all patients aged ≥40 years who regularly visited the Marumori Hospital Department of Internal medicine between 1 September 2012 and 31 August 2013. Outpatient physicians at Marumori Hospital explained the need for CRC screening and recommended FOBT for all patients who reported that they were untested; the physicians then conducted FOBT for patients who requested it. Secondary testing was recommended for patients with positive FOBT results. At this point, only patients who provided oral consent for participation in the study and for the future examination of data were included as participants. For patients who did not or could not provide stool samples after providing consent for FOBT, we evaluated their reasons for not submitting the samples.

### 2.2. Methods

We prospectively collected the following data from patients with positive FOBT results: age; sex; comorbidities; history of antithrombotic agent and non-steroidal anti-inflammatory drug (NSAID) use; whether they underwent secondary testing and the respective test results; and whether they underwent upper gastrointestinal (GI) endoscopy and the respective results. We investigated the reasons for patients who did not submit their stool samples. The present study is a retrospective cohort study of FOBT-positive patients; we referred to the medical records as of February 2021 to assess the outcomes. If the cause of death was unclear in the medical records, we confirmed it with the patient’s family.

FOBTs employ an immunological method, i.e., an immunochromatography assay; it is used to identify human hemoglobin and transferrin in fecal specimens using antibody-antigen reaction; it is more sensitive than chemical methods [13]. In a qualitative assessment with OC-Hemocatch S (Eiken Chemical; Tokyo, Japan) [14], we defined ≥1 positive result in a 2-day sampling method as positive for fecal occult blood. Antithrombotic agents included antiplatelet drugs (enteric-coated aspirin tablets, ticlopidine, clopidogrel, cilostazol) and anticoagulants (warfarin potassium, dabigatran) that could be prescribed anytime during FOBT. Participants who used any of the above drugs regularly at least 14 days prior to FOBT were considered to be “on medication.”

Secondary testing principally consisted of colonoscopy. However, when colonoscopy was difficult to perform because of reasons such as reduction in activities of daily living, abdominal computed tomography (CT) was used. A positive result in colonoscopy was defined based on colonoscopy screening and surveillance guidelines (Japan Gastroenterological Endoscopy Society) [15] for cases wherein surveillance colonoscopy was recommended within 3 years, i.e., (1) ≥3 polyps measuring < 10 mm; (2) ≥1 polyp measuring ≥ 10 mm; or (3) histopathological diagnosis of tubulovillous adenoma, high-grade dysplasia, or carcinoma. In abdominal CT, patients who demonstrated intestinal masses distinguishable from surrounding fecal masses were considered to show positive results, while the other patients were considered to have negative results. To assess the presence of malignancies from the oral cavity to the upper GI tract, FOBT-positive patients who provided consent also underwent an upper GI endoscopy.

We calculated the FOBT-positive rate and CRC detection rate in the present study and compared them with the CRC test results compiled by Marumori town in 2012 [16], along with the nationwide data for GI cancer in 2012 [17], i.e., nearly the same period as that of the present study. To adjust the comparisons for age, we abstracted the data for patients aged ≥ 65 years in the nationwide cohort and conducted an additional analysis. Using the Kaplan–Meier method, we calculated overall survival (OS) and disease-specific survival (DSS) among the FOBT-positive patients based on mortality data as of February 2021. We also described the upper GI endoscopy results for patients who underwent this examination. Lastly, we analyzed the reasons regarding the failure to submit stool samples for FOBT after the patients provided consent to undergo testing and to participate in the study.

When we explained FOBT to all participants, we also explained that the test results and other data may be used following linkable anonymization; not participating in the study would confer no disadvantage; and the participants could withdraw from the study at any time. We used data only from the patients who provided consent after the above explanation.

### 2.3. Ethical Considerations

The present study involved the use of patients’ personal information; thus, we aimed to protect this information in accordance with the Declaration of Helsinki. We analyzed and assessed patients’ data only after obtaining approval from the Marumori Hospital Institutional Review Board (approval no. IRB 2021-1). We published the title of the study on the hospital’s website and provided participants an opportunity to opt out by displaying the details of the study at a conspicuous location.

### 2.4. Statistical Processing

Statistical analyses were performed using R version 4.1.1 and JMP^®^ pro 16. Nominal variables were compared using the Fisher’s exact test; survival curves were drawn using the Kaplan–Meier method; relative risks and confidence intervals were calculated using logistic regression analysis. The level of statistical significance was defined as *p* < 0.05.

## 3. Results

Of the 425 patients who did not undergo CRC screening in the previous two years, 420 provided consent for FOBT and for participation in the present study. In the intention-to-treat analysis, which included patients who did not provide consent, 388 of 425 patients (91%) provided stool samples. In total, 32 patients did not submit stool samples after consenting to participate; the reasons for this included dementia or other problems with memory (n = 9, 24%) and physical difficulty in collecting samples (n = 9, 24%). Of the 388 samples submitted, 66 (17%) were positive for fecal occult blood (Figure 1).

Of the 66 FOBT-positive patients (mean age, 73 years), 35 were men, 15 used antithrombotic agents, and 12 used NSAIDs. Of these 66 patients, 61 (92%) had comorbid lifestyle disease, cardiovascular disease, or chronic respiratory disease, while 32 (48%) had two or more comorbidities. Fifty-six of the sixty-six FOBT-positive patients requested secondary testing. Colonoscopy and abdominal CT were performed in 52 and 4 patients, respectively, and the secondary tests showed positive results in 22 patients (39%), of which 3 patients showed CRC. Thus, CRC was detected in 3 of the 388 patients (0.77%) who provided stool samples. One of these patients achieved complete remission with endoscopic mucosal resection, one achieved complete remission with surgical resection, and the other underwent palliative care, following diagnosis of advanced CRC with lymph node metastasis at the time of abdominal CT (Table 1).

The FOBT-positive rate in the present study (17%) was significantly higher than that during medical checkups for residents of Marumori town (6.0%; *p* < 0.001) and in the nationwide data (5.7%; *p* < 0.001). The CRC detection rate in the present study (0.77%) did not differ significantly from the data for Marumori (0.26%; *p* = 0.18), but was significantly higher than the nationwide data (0.13%; *p* < 0.05) (Figure 2). In a sensitivity analysis, a comparison with the nationwide data for the population of patients aged ≥65 years (after age adjustment of the study population) revealed that while the FOBT-positive rate was significantly higher in the present study, the nationwide CRC detection rate in patients aged ≥65 years did not differ significantly from the findings of the present study.

As of 16 February 2021, 66 of the FOBT-positive patients were confirmed to have died. The causes of death were verified using medical records for 12 patients and via phone calls with the families for the remaining 4 patients. We calculated the OS and DSS (i.e., survival specific to CRC) following positive FOBT results for the 66 fecal occult blood-positive subjects. At 5 years and 7 years, OS was 87.1% (95% CI: 76.3–93.5%) and 74.7% (95% CI: 62.1–84.2%), respectively, while DSS was 98.5% at both points (95% CI: 89.9–99.8%). Although the most common cause of death was cancer (n = 6), CRC-related death was recorded in only one case, wherein CRC was detected during the current project and involvement of lymph node metastasis was noted in secondary abdominal CT (Figure 3).

Of the 56 patients who underwent secondary testing, 39 (70%) also underwent upper GI endoscopy. Except for one patient who showed positive findings in secondary testing and demonstrated early-stage esophageal cancer, no other patients demonstrated any malignancies.

## 4. Discussion

We used a retrospective cohort to obtain details regarding patients who regularly visited a regional primary care facility and were not tested for CRC in the previous two years; we also aimed to determine the long-term outcomes in patients who tested positive for fecal occult blood. The first important finding of the study is that 388 of 425 unscreened patients (91%) understood the importance of CRC screening and managed to submit stool samples based on an explanation from an outpatient physician. We surmised that problems related to both patients and medical staff would have influenced stool sample collection; patients may have had a cognitive bias, i.e., they did not feel the need to undergo CRC screening because they were already visiting a hospital, although these were regular visits for diseases unrelated to CRC. The medical staff did not devote enough time to confirm whether patients had undergone CRC screening or to explain the need for screenings in daily outpatient practice. We believe this is precisely why explanations by outpatient physicians regarding the need for appropriate screening led to over 90% of untested subjects undergoing screening. FOBT is simple, and primary care physicians can easily check whether their patients have undergone CRC screening and can proactively conduct tests.

Patients who regularly visited Marumori Hospital but were not tested for CRC had a mean age of 73 years, and 48% of these patients had multiple comorbidities. Among the patients who submitted stool samples to Marumori Hospital, the FOBT-positive rate (17%) and the CRC detection rate (0.77%) were significantly higher than the corresponding nationwide data. In the nationwide data, the majority of patients were aged around 50 to 60 years [17]; the patients in the present study were older than those in the nationwide data, which could explain our results. In fact, when we abstracted the findings of patients aged ≥65 years from the nationwide data, we did not observe significant differences with the CRC detection rate in the present study. Thus, the considerable number of patients who regularly visited a regional medical center but were not tested for CRC had at least the same risk of CRC as people of the same age group that were tested, suggesting that identification of unscreened patients is important. A study from outside Japan showed that patients with CRC who have multiple comorbidities are at a higher risk of mortality than patients without comorbidities [18]. Thus, failure in detecting CRC when patients visit a primary care medical center for non-related diseases (such as lifestyle diseases) puts them at high-risk of CRC.

Of the 66 FOBT-positive patients in the present study whose outcomes could be followed up for 7 years (including 10 patients who did not request secondary testing), except for 1 patient who already had advanced CRC with lymph node metastasis at the time of secondary testing, no CRC-related deaths were recorded during the follow-up period. In a previous study, the 5-year survival rate for advanced CRC in unscreened patients was 23.4% lower than that in patients who were screened (66.7% vs. 90.1%) [19]. Considering the two patients in the present study with newly-detected CRC (who consequently achieved complete remission), the present project could help in improving the survival rate for patients that are not tested for CRC. The 7-year DSS in the present study was 98.5%; thus, conducting FOBT even only once every couple of years and recommending secondary testing for patients who test positive may reduce the rates of CRC-related death for several years.

Considering studies from outside Japan, there are conflicting results regarding the effectiveness of upper GI endoscopy among patients who test positive for fecal occult blood but reveal no significant findings in colonoscopy [20,21]. In the present study, there were no significant findings in upper GI endoscopy among the patients who showed negative findings in the colonoscopy. However, the prevalence of gastric cancer is considered high in Japan; thus, additional upper GI endoscopy for patients with negative findings in the colonoscopy may be effective.

Lastly, many participants did not or could not submit stool samples, despite providing consent to participate in the current project because of impairment in cognitive function or physical limitations caused by problems such as sequelae of cerebral infarction (Figure 1). The same could be true for people who do not undergo CRC screening. Providing high-quality screening examinations that account for individual risk is especially important in rural areas with few medical resources [8]. Assisting patients with dementia or physical disabilities in collecting stool samples may facilitate more CRC screening and allow early detection of CRC.

One limitation of the present study is the small sample size because this was a single-center study. In the present study, two participants underwent treatment for CRC, while one participant died; a larger-scale, multicenter study is necessary to determine the impact of CRC screening on mortality reduction. Moreover, the results of the present study may not be generalizable in populations with a different age distribution of outpatients, such as at medical centers in urban areas, where patients are relatively younger. Additionally, the present study involved a retrospective cohort, which prevented us from obtaining information about detailed CRC-related risk factors, such as diet and genetic background. Furthermore, FOBT-negative patients were not listed at the beginning of the study, hindering direct comparisons between FOBT-positive and -negative patients. Including information on FOBT-negative patients could have improved the quality of our analysis; thus, future prospective cohort studies may yield novel findings by considering information for FOBT-negative patients. However, no study has compared cancer-related mortality with FOBT-negative patients in general cancer screening. Therefore, we believe that the present study’s comparisons of FOBT-positive rates and CRC detection rates with a single-town and nationwide data during the same period are acceptable.

Nevertheless, a higher CRC detection rate than that in nationwide figures and the novel detection of CRC in three patients in an area roughly comprising 15,000 people demonstrate that the use of simple FOBT at regional medical centers is highly meaningful. While complete understanding of the risk of CRC among outpatients is likely uncommon in the real world, we believe that the present study provided results that are applicable in real-world clinical practice. In future, conducting similar studies prospectively at multiple centers can further elucidate the risk of CRC among patients who have not undergone screening. 

## 5. Conclusions

Proactive FOBT recommendations led to FOBT performance in over 90% of regular outpatients and patients who were not screened for CRC; it also enabled detection of treatable CRC in two patients. Regular outpatients at medical centers that are often elderly and have comorbidities may have at least the same risk of CRC as that in patients aged ≥65 years who have undergone screening examinations.

## Figures and Tables

**Figure 1 diagnostics-12-01782-f001:**
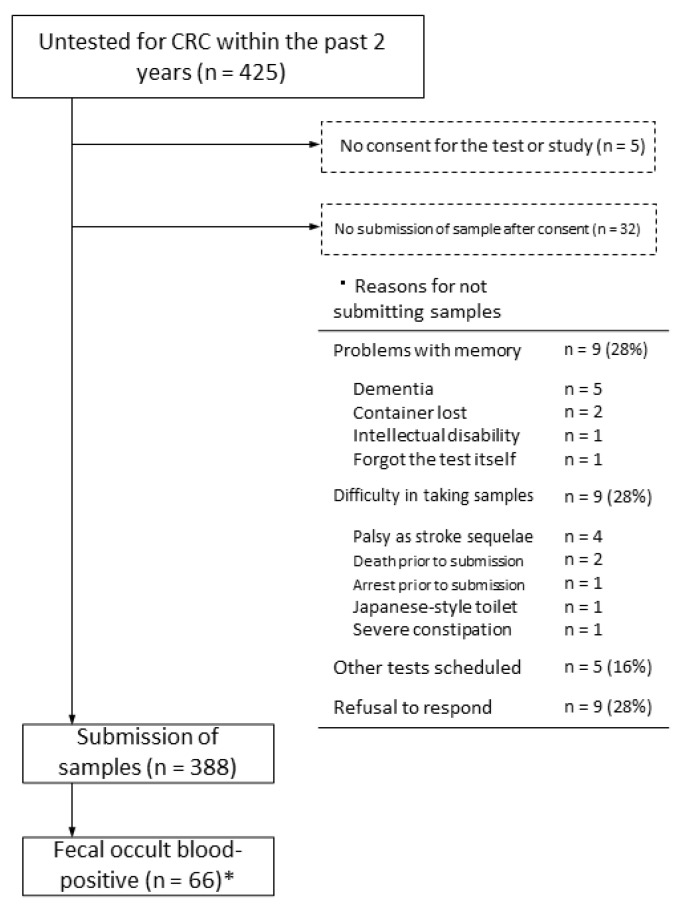
Flowchart of patients who were untested for CRC up to secondary testing. * Cohort study participants.

**Figure 2 diagnostics-12-01782-f002:**
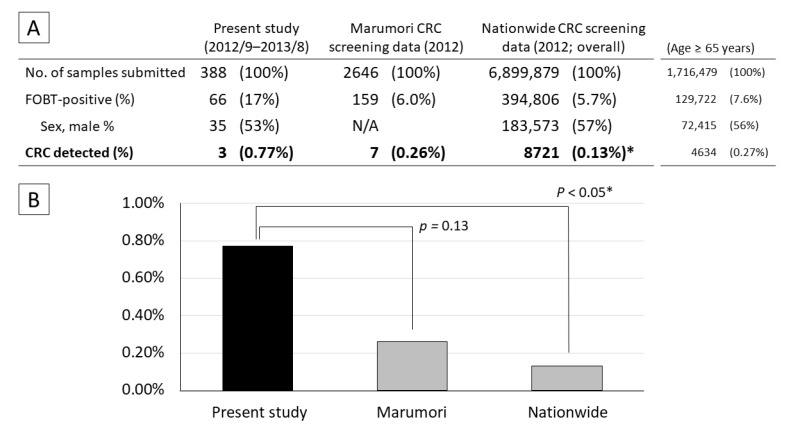
Comparison of CRC screening results in the present study with those of Marumori town and nationwide data. (**A**) Number of patients who submitted stool samples for FOBT and tested positive. (**B**) CRC detection rates. The data for Marumori and all of Japan were obtained from a cancer screening project; they do not include patients who did not undergo screening; therefore, they do not overlap with the present study completely. Fisher’s exact test: * vs. present study; *p* < 0.05.

**Figure 3 diagnostics-12-01782-f003:**
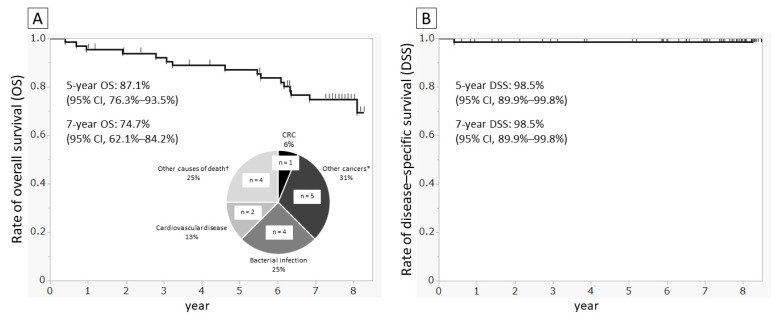
Survival curves and causes of death. Curves drawn using the Kaplan–Meier method. (**A**) Overall survival and causes of death and (**B**) disease-specific survival for CRC in 66 fecal occult blood-positive patients among FOBT submission until February 2021. * Lung cancer in two patients and one duodenal papillary carcinoma, one prostate cancer, and one renal carcinoma. ^†^ One patient with heart failure, one with failure to thrive, one with drowning, and one with sudden death during hospitalization.

**Table 1 diagnostics-12-01782-t001:** Characteristics of fecal occult blood-positive patients.

	Positive Patients (n = 66)
Mean age, years (±SD)	73 (±9.6)
Sex, male (%)	35 (53)
Antithrombotic agent use, n (%) *	15 (23)
NSAID use, n (%)	12 (18)
Comorbidities, n (%)	59 (89)
Hypertension	44 (67)
Hyperlipidemia	21 (32)
Diabetes	15 (23)
Old cerebral infarction	7 (11)
Chronic atrial fibrillation	6 (9.1)
Cardiovascular disease	6 (9.1)
Chronic respiratory disease	4 (6.1)
Multiple comorbidities, n (%)	32 (48)
Secondary testing, n (%) ^†^	56 (85)
Positive for findings, n (% of patients who underwent secondary testing)	22 (39)
Carcinoma, n ^‡^	3
Other polyps ^§^	19

SD: Standard deviation. NSAIDs: Non-steroidal anti-inflammatory drugs. * Includes antiplatelet drugs (enteric-coated aspirin tablets, ticlopidine, clopidogrel, cilostazol) and anticoagulants (warfarin potassium, dabigatran). ^†^ Colonoscopy (n = 52), plain abdominal CT (n = 3), abdominal CT with contrast (n = 1). ^‡^ Adenocarcinoma diagnosed on the basis of endoscopic mucosal dissection (n = 1), adenocarcinoma diagnosed on the basis of surgical resection (n = 1), CRC with remote lymph node metastasis diagnosed on the basis of abdominal CT (n = 1). ^§^ Tubular adenoma (high grade) (n = 1), tubular adenoma (low grade) (n = 15), tubulovillous adenoma (n = 1), no biopsy performed (n = 2).

## Data Availability

Not applicable.

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
