# Peer review of "Feasibility of Support by Family Practitioners in Reducing Colorectal Cancer-Related Death among Outpatients Who Have Not Undergone Colorectal Cancer Screening"

_diagnostics, 2022, doi:10.3390/diagnostics12081782_

Round 1
Reviewer 1 Report
A brave amount of English language is required, assume that your abstract was written last and not passed by language check (as an indicator)
Author Response
Thank you for your comment on our English writing, especially on the abstract. We asked for additional native checks before submitting the revised manuscript. Changes due to native checks are colored blue in the attached file and the revised manuscript, and most of them were in the abstract section.
Abstract: We aimed to clarify the effectiveness of interventions in outpatients who did not undergo colorectal cancer (CRC) screening. From September 2012 to August 2013, we conducted a project, which showed that the immunological fecal occult blood test (FOBT) was actively recommended for outpatients who were ≥40 years of age, attended the Marumori Hospital regularly, and were not screened for CRC in the previous two years. We evaluated the detection rate of CRC and the disease specific survival ratio in February 2021 among patients with positive FOBT results during the retrospective cohort study. Overall, 388 (91%) out of the 425 outpatients submitted their stool samples. Among 388 outpatients, 66 tested positive for FOBT. While both the positive rate of FOBT and the detection rate of CRC (17% and 0.77%, respectively) were significantly higher than those in the nationwide administrative examination (5.7% and 0.13%, respectively) (p<0.05), there was no statistically significant difference in the detection rate comparing with the group aged 65 years and older in the nationwide administrative examination. The 7-year CRC-specific survival ratio was 98.5%. Active promotion of FOBT at primary care institutions for outpatients who did not undergo CRC screening may contribute to reducing the frequency of CRC-related deaths.

Reviewer 2 Report
This is a description of a project where FOBT was offered to people attending a hospital.
1. The title is misleading as this is not a primary practice project and the authors are not family physicians.
2. No conclusion about the efficacy of the approach can be made without further knowledge about the patient profile of the tested population.
3. Apparently, the population contained many people with severe and combined comorbidities and many with dementia, and it it questionable and unproven whether cancer screening for these individuals is beneficial and recommended.
Author Response
Reviewer 2
Comments and Suggestions for Authors
This is a description of a project where FOBT was offered to people attending a hospital.
- The title is misleading as this is not a primary practice project and the authors are not family physicians.
Respond 1:
I’m sorry for misunderstanding. First, Marumori hospital is a primary care institution in Marumori town. In Japan, there are many hospitals playing roles as both primary care and inpatient management of common disease, especially in rural area. Second, five of six authors (R.S, T. Kanno, Y.H, M.M, and M.O) were affiliated with Marumori Hospital as primary care physicians during from the study enrollment period to the end of follow-up. Then the one of the authors (T Koike) joined as a supervisor for this research.
To avoid misleading, we would state clearly about above description in the main text.
1. Introduction, paragraph 3, line 51 “To address this objective, we developed a project at Marumori Hospital, a regional primary care hospital in Marumori…”
Author contributions, P.8, line302:”Five of the six authors (R.S, T. Kanno, Y.H, M.M, and M.O) were affiliated with Marumori Hospital as primary care physicians from the study-enrollment period to the end of follow-up.”
- No conclusion about the efficacy of the approach can be made without further knowledge about the patient profile of the tested population.
Respond 2:
We appreciate your indication. We know the shortcomings of this study in that we were not able to collect patient profiles as adequately as in a prospective study. Therefore, we stated and discussed about the issue in the limitation section as” Furthermore, FOBT-negative patients were not listed at the beginning of the study, hindering direct comparisons between FOBT-positive and -negative patients. Including information on FOBT-negative patients could have improved the quality of our analysis; thus, future prospective cohort studies may yield novel findings by considering information for FOBT-negative patients. However, no study has compared cancer-related mortality with FOBT-negative patients in general cancer screening. Therefore, we believe that the present study’s comparisons of FOBT-positive rates and CRC detection rates with a single-town and nationwide data during the same period are acceptable.” (4. Discussion, paragraph 6, P.8, line 273). And we added some words to help understanding this description. Even with this limited approach, we believe it is possible to conclude that active colorectal screening in hospitals with many elderly outpatients can be recommended.
- Apparently, the population contained many people with severe and combined comorbidities and many with dementia, and it it questionable and unproven whether cancer screening for these individuals is beneficial and recommended.
Respond 3:
Thank you for your indication. The reviewer probably concerned risk of patients with severe dementia. As the reviewer mentioned, the effectiveness of cancer screening may be limited in patients with severe dementia who are unable to make their own treatment decisions. We were not able to prospectively check the prevalence of dementia in participants in this retrospective research. On the other hand, the epidemiological studies in Japan have reported that the prevalence of dementia is about 5-10% in people in their 70s, which is the average age of the participants in this study. Additionally, there have been several reports recently that aggressive early detection of gastrointestinal cancer patients over the age of 75 and even 85 may help maintain healthy life expectancy. As a result, we believe that early detection of colorectal cancer is useful because the participants in this study were excluded inpatients and all of them had ADL that allowed them to go to the hospital as outpatients, and even if about 10% of them may have dementia. We thank you for taking the time to peer review.

Reviewer 3 Report
This is an interesting reprot with subborden evidence. I recommend it can be published with some english improve.
Author Response
Reviewer 3
Comments and Suggestions for Authors
This is an interesting reprot with subborden evidence. I recommend it can be published with some english improve.
Respond:
Thank you for reviewing our manuscript. We believe that this article could help for family practitioners.
Reviewer 4 Report
In the article “ Feasibility of support by family practitioners in reducing colorectal cancer-related death among outpatients who have not undergone colorectal cancer screening” the authors applied immunochromatography as a diagnostic technique to retrospectively assess fecal occult blood tests (FOBT) in patients (n=425). The positive FOBT rate in the present study (n=66; 17%) was significantly higher than that of patients scheduled for medical controls. Overall, the article is well written, with an adequate description of the methodology and convincing results (an adequate statistical analysis is presented). This preliminary study highlights the usefulness of an immunological diagnostic method (immunochromatography) for colorectal cancer screening, which is relevant for the opportune detection of CRC.
As a minor comment, I suggest the following: 1. include technical details of the diagnostic procedure by immunochromatography, and 2. improve the image quality of figure 1.
Author Response
Reviewer 4
Comments and Suggestions for Authors
In the article “ Feasibility of support by family practitioners in reducing colorectal cancer-related death among outpatients who have not undergone colorectal cancer screening” the authors applied immunochromatography as a diagnostic technique to retrospectively assess fecal occult blood tests (FOBT) in patients (n=425). The positive FOBT rate in the present study (n=66; 17%) was significantly higher than that of patients scheduled for medical controls. Overall, the article is well written, with an adequate description of the methodology and convincing results (an adequate statistical analysis is presented). This preliminary study highlights the usefulness of an immunological diagnostic method (immunochromatography) for colorectal cancer screening, which is relevant for the opportune detection of CRC.
As a minor comment, I suggest the following: 1. include technical details of the diagnostic procedure by immunochromatography, and 2. improve the image quality of figure 1.
Respond:
Thank you for your suggestions. We added the description of technical details of the diagnostic procedure by immunochromatography as below and improved the image quality of figure 1.
2.2 Methods - paragraph2, page 2, line 80: FOBTs employ an immunological method, i.e., an immunochromatography assay; it is used to identify human hemoglobin and transferrin in fecal specimens using antibody-antigen reaction; it is more sensitive than chemical methods [13].
Round 2
Reviewer 1 Report
there are still some language issues before this becomes acceptable for presentation.
Reviewer 2 Report
The authors have modified the manuscript satisfactorily.